# [20(22)*E*]-Lanostane Triterpenes from the Fungus *Ganoderma australe*

**DOI:** 10.3390/jof8050503

**Published:** 2022-05-12

**Authors:** Lin Zhou, Li-Li Guo, Masahiko Isaka, Zheng-Hui Li, He-Ping Chen

**Affiliations:** 1School of Pharmaceutical Sciences, South-Central Minzu University, Wuhan 430074, China; 2019110443@mail.scuec.edu.cn (L.Z.); g15513998615@163.com (L.-L.G.); 2National Center for Genetic Engineering and Biotechnology (BIOTEC), 113 Thailand Science Park, Phaholyothin Road, Klong Luang, Pathumthani 12120, Thailand; isaka@biotec.or.th

**Keywords:** *Ganoderma australe*, triterpene, 20(22)*E* configuration, PGME method

## Abstract

Twelve new lanostane triterpenoids (**1**–**5**, **7**–**13**) were isolated from the fruiting bodies of the fungus *Ganoderma australe*. The structures of the new compounds were elucidated by extensive 1D and 2D NMR, and HRESIMS spectroscopic analysis. All the triterpenes are featured by 20(22)*E* configurations which are uncommon in the *Ganoderma* triterpene family. The absolute configuration of the C-25 of compounds **1**, **2**, and **6** were determined by the phenylglycine methyl ester (PGME) method. A postulated biosynthetic pathway for compound **1** was discussed. This study opens new insights into the secondary metabolites of the chemically underinvestigated fungus *G. australe*.

## 1. Introduction

Mushrooms are popular in the food market due to their delicious taste and nutrition values. Mushroom-derived secondary metabolites have contributed lots of lead compounds for medical and agricultural use. Psilocybin, a specialized compound from the genus *Psilocybe*, is a naturally occurring hallucinogenic prodrug for treating psychiatric disorders [1]. Strobilurins, firstly originated from the mushroom *Strobilurus tenacellus*, are a group of natural products and their synthetic analogs are used in agriculture as fungicides [2,3]. More and more attention has been paid to mining promising lead compounds from the mushroom natural product reservoir in recent years.

*Ganoderma*, called “*lingzhi*” in China, is a group of wood-decaying mushrooms with hard fruiting bodies which grow mostly in spare scatting sunshine, on the trees, and on open grounds [4]. It is a genus of notable medicinal fungi and traditional herbal medicine for the treatment of diseases such as hepatopathy, nephritis, neurasthenia, and asthma [5,6,7,8,9,10]. The *Shennong Ben Cao Jing*, an ancient Chinese medicinal book, documented that *Ganoderma* was effective for maintaining health, prolonging life, boosting memory, and relieving stress. *Ganoderma lucidum* and *G. sinense* are two registered species recorded in the *Chinese Pharmacopoeia* (2015). Many studies show that triterpenoids and polysaccharides are the main bioactive substances in *Ganoderma* [11,12,13,14,15,16,17]. *Ganoderma australe* is a species used in folk medicine as the alternative of *G. lucidum*. However, this fungus has rarely been chemically investigated compared to other *Ganoderma* species, such as *G. lucidum*, *G. cochlear*, and *G. sinense*. Previous studies on this fungus have led to the isolation of lanostane triterpenes [18,19,20,21], meroterpenoids [22,23], and alkaloids [22]. The lanostanoids from this species are over-oxygenated compared to the ones isolated from other species of *Ganoderma*, especially the position of C-20 [18,20]. The quaternary hydroxy group substituted at C-20 led to the introduction of an additional chiral carbon of which the stereochemistry was difficult to be assigned even by chemical derivatization. Moreover, this substituted pattern of the C-20 hydroxy group always triggered to dehydration between C-21 to produce the 20(22)-double bond, which always incorporated into an α,β-unsaturated ketone group with the C-23 carbonyl group. In this study, we have investigated the secondary metabolite profiles of *G. australe*, which led to the isolation of twelve new highly oxygenated lanostane triterpenes with uncommon 20(22)*E* configurations. We, herein, report the isolation and structural elucidation of the new compounds.

## 2. Experimental Section

### 2.1. General Experimental Procedures

Optical rotations were obtained on an Autopol IV-T digital polarimeter (Rudolph, Hackettstown, NJ, USA). UV spectra were recorded on a Hitachi UH5300 spectrophotometer (Hitachi, Tokyo, Japan). CD spectra were measured on a Chirascan Circular Dichroism spectrometer (Applied Photophysics Limited, Leatherhead, Surrey, UK). The 1D and 2D spectra were obtained on the Bruker Avance III 500 MHz and 600 MHz spectrometers (Bruker Corporation, Karlsruhe, Germany). HRESIMS spectra were measured on a Q Exactive Orbitrap mass spectrometer (Thermo Fisher Scientific, Waltham, MA, USA). Single-crystal X-ray diffraction data were recorded on the BRUKER D8 QUEST diffractometer (Bruker Corporation, Karlsruhe, Germany). Medium pressure liquid chromatography (MPLC) was performed on an Interchim system equipped with a column packed with RP-18 gel (40–75 μm, Fuji Silysia Chemical Co., Ltd., Kasugai, Japan). Preparative high performance liquid chromatography (prep-HPLC) was performed on an Agilent 1260 Infinity Ⅱ liquid chromatography system equipped with a Zorbax SB-C_18_ column (particle size 5 μm, dimensions 150 mm × i.d. 9.4 mm, flow rate 5 mL⋅min^−1^) and a DAD detector (Agilent Technologies, Santa Clara, CA, US). Sephadex LH-20 (GE Healthcare, Stockholm, Sweden) and silica gel (200–300 mesh, Qingdao Haiyang Chemical Co., Ltd., Qingdao, China) were used for column chromatography (CC). (*S*)- and (*R*)-phenylglycine methyl ester were bought from Sigma-Aldrich (St. Louis, MO, USA).

### 2.2. Fungal Material

The fruiting bodies of *G. australe* were collected in Tongbiguan Natural Reserve, Dehong, Yunnan Province, China, in 2016, and identified by Prof. Yu-Cheng Dai (Institute of Microbiology, Beijing Forestry University). A voucher specimen of *G. australe* was deposited in the Mushroom Bioactive Natural Products Research Group of South-Central University for Nationalities.

### 2.3. Extraction and Isolation

The dry fruiting bodies of *G. australe* (3.26 kg) were extracted four times with CHCl_3_:MeOH (1:1) at room temperature to obtain crude extract, which was further suspended in distilled water and partitioned against EtOAc to afford EtOAc layer extract (130 g). The EtOAc layer extract was eluted on MPLC with a stepwise gradient of MeOH−H_2_O (20:80−100:0) to afford eight fractions (A−H).

Fraction C was subjected to Sephadex LH-20 (MeOH) and obtained 16 subfractions (C1-C16), and C2 was separated by prep-HPLC (MeCN−H_2_O: 20:80−40:60, 25 min, 4 mL/min) to yield compound **2** (6.4 mg, t_R_ = 20.5 min).

Fraction D was separated by Sephadex LH-20 (MeOH) to give eight subfractions (D1–D8). Subfraction D4 was subjected to silica gel CC (petroleum ether−acetone from *v/v* 6:1 to 1:1) and yielded eleven subfractions (D4a–D4k). Subfraction D4d was purified by prep-HPLC (MeCN−H_2_O: 20:80–40:60, 25 min, 4 mL/min) to yield compound **3** (2.0 mg, t_R_ = 19.0 min).

Fraction E was separated by Sephadex LH-20 (CHCl_3_:MeOH = 1:1) to afford four subfractions (E1–E4). E2 was separated by CC on silica gel (petroleum ether−acetone from *v/v* 15:1 to 1:1) to obtain 10 subfractions (E2a–E2j). E2b was subjected to prep-HPLC (MeCN–H_2_O: 70:30–90:10, 25.0 min, 4 mL/min) to obtain compound **7** (3.8 mg, t_R_ = 14.0 min) and **8** (3.7 mg, t_R_ = 15.0 min). Compound **10** (3.7 mg, t_R_ = 21.0 min) was purified from E2f by prep-HPLC (MeCN–H_2_O: 70:30–90:10, 25 min, 4 mL/min). E4 was subjected to CC on silica gel (petroleum ether–acetone from *v/v* 10:1 to 1:1) to obtain 12 fractions. Compound **12** (6.0 mg, t_R_ = 18.0 min) was purified from E4c by prep-HPLC (MeCN–H_2_O: 40:60–60:80, 25 min, 4 mL/min). Compound **6** (5.1 mg, t_R_ = 19.1 min) was purified from E4d by prep-HPLC (MeCN–H_2_O: 40:60–60:80, 25.2 min, 4 mL/min). Compound **9** (21.4 mg, t*_R_* = 20.0 min) was purified from E4f by prep-HPLC (MeCN–H_2_O: 40:60–60:80, 25 min, 4 mL/min). Compound **5** (7.3 mg, t_R_ = 30.0 min) was purified from E4g by prep-HPLC (MeCN–H_2_O: 45:55–73:27, 35.0 min, 4 mL/min). Compound **11** (2.6 mg, t_R_ = 27.0 min) was purified from E4h by prep-HPLC (MeCN–MeOH–H_2_O: 40:20:40, isocratic, 30 min, 4 mL/min). E7 was subjected to column chromatography (CC) on silica gel (petroleum ether–acetone from *v/v* 15:1 to 1:1) to obtain 13 fractions. E7j was purified by prep-HPLC (MeCN–H_2_O: 30:70–50:50, 25 min, 4 mL/min) to yield compound **1** (8.6 mg, t_R_ = 14.0 min).

Fraction G was separated by column chromatography (CC) on silica gel (petroleum ether–acetone from *v/v* 15:1 to 1:1) to obtain 10 subfractions (G1–G10). Compound **4** (13.3 mg, t_R_ = 12.7 min) was purified from G7 by prep-HPLC (MeCN–H_2_O: 50:50–30:70, 25 min, 4 mL/min). Compound **13** (4.0 mg, t_R_ = 13.2 min) was purified from G7 by prep-HPLC (MeCN–H_2_O: 50:50–30:70, 25 min, 4 mL/min).

### 2.4. Spectroscopic Data

Ganoaustralenone A (**1**): yellow oil. [*α*]${}_{D}^{24}$ +8.9 (*c* 0.09, MeOH); UV (MeOH) λ_max_ (log *ε*) 250.0 (4.22); ^1^H NMR (600 MHz, CD_3_OD) data, see Table 1, ^13^C NMR (150 MHz, CD_3_OD) data, see Table 2 HRESIMS *m/z* 535.26685 [M + Na]^+^ (calcd for C_30_H_40_O_7_Na, 535.26717).

Ganoaustralenone B (**2**): white powder. [*α*]${}_{D}^{24}$ +81.3 (*c* 0.06, MeOH); UV (MeOH) λ_max_ (log *ε*) 250.0 (4.00); ^1^H NMR (600 MHz, CDCl_3_) data, see Table 1, ^13^C NMR (150 MHz, CDCl_3_) data, see Table 2; HRESIMS *m/z* 551.26422 [M + Na]^+^ (calcd for C_30_H_40_O_8_Na, 551.26209).

Ganoaustralenone C (**3**): white powder. [*α*]${}_{D}^{24}$ +178.4 (*c* 0.05, MeOH); UV (MeOH) λ_max_ (log *ε*) 250.0 (4.20); ^1^H NMR (500 MHz, CDCl_3_) data, see Table 1, ^13^C NMR (125 MHz, CDCl_3_) data, see Table 2; HRESIMS *m/z* 537.28180 [M + Na]^+^ (calcd for C_30_H_42_O_7_Na, 537.28227).

Ganoaustralenone D (**4**): yellow oil. [*α*]${}_{D}^{24}$ +102.2 (*c* 0.13, MeOH); UV (MeOH) λ_max_ (log *ε*) 250.0 (4.24); ^1^H NMR (600 MHz, CD_3_COCD_3_) data, see Table 1, ^13^C NMR (150 MHz, CD_3_COCD_3_) data, see Table 2; HRESIMS *m/z* 565.27728 [M + Na]^+^ (calcd for C_31_H_42_O_8_Na, 565.27774).

Ganoaustralenone E (**5**): yellow oil. [*α*]${}_{D}^{24}$ +25.3 (*c* 0.07, MeOH); UV (MeOH) λ_max_ (log *ε*) 250.0 (4.27); ^1^H NMR (600 MHz, CDCl_3_) data, see Table 1, ^13^C NMR (150 MHz, CDCl_3_) data, see Table 2; HRESIMS *m/z* 549.28210 [M + Na]^+^ (calcd for C_31_H_42_O_7_Na, 549.28282).

Ganoaustralenone F (**7**): yellow oil. [*α*]${}_{D}^{24}$ +5.7 (*c* 0.04, MeOH); UV (MeOH) λ_max_ (log *ε*) 245.0 (3.53); ^1^H NMR (800 MHz, CDCl_3_) data, see Table 3, ^13^C NMR (150 MHz, CDCl_3_) data, see Table 2; HRESIMS *m/z* 547.30286 [M + Na]^+^ (calcd for C_32_H_44_O_6_Na, 547.30356).

Ganoaustralenone G (**8**): yellow oil. [*α*]${}_{D}^{24}$ +17.1 (*c* 0.04, MeOH); UV (MeOH) λ_max_ (log *ε*) 245.0 (3.62); ^1^H NMR (800 MHz, CDCl_3_) data, see Table 3, ^13^C NMR (200 MHz, CDCl_3_) data, see Table 4; HRESIMS *m/z* 561.28223 [M + Na]^+^ (calcd for C_32_H_42_O_7_Na, 561.28282).

Ganoaustralenone H (**9**): yellow oil. [*α*]${}_{D}^{24}$ +31.3 (*c* 0.11, MeOH); UV (MeOH) λ_max_ (log *ε*) 250.0 (3.39); ^1^H NMR (600 MHz, CDCl_3_) data, see Table 3, ^13^C NMR (150 MHz, CDCl_3_) data, see Table 4; HRESIMS *m/z* 513.32135 [M + H]^+^ (calcd for C_31_H_45_O_6_, 513.32161).

Ganoaustralenone I (**10**): pale yellow oil. [*α*]${}_{D}^{24}$ +67.7 (*c* 0.05, MeOH); UV (MeOH) λ_max_ (log *ε*) 245.0 (4.02); ^1^H NMR (600 MHz, CDCl_3_) data, see Table 3, ^13^C NMR (150 MHz, CDCl_3_) data, see Table 4; HRESIMS *m/z* 563.29749 [M + Na]^+^ (calcd for C_32_H_44_O_7_Na, 563.29847).

Ganoaustralenone J (**11**): yellow oil. [*α*]${}_{D}^{24}$ +119.56 (*c* 0.05, MeOH); UV (MeOH) λ_max_ (log *ε*) 245.0 (4.17); ^1^H NMR (600 MHz, CDCl_3_) data, see Table 5, ^13^C NMR (150 MHz, CDCl_3_) data, see Table 4; HRESIMS *m/z* 549.28210 [M + Na]^+^ (calcd for C_31_H_42_O_7_Na, 549.28282).

Ganoaustralenone K (**12**): pale yellow oil. [*α*]${}_{D}^{24}$ +149.2 (*c* 0.06, MeOH); UV (MeOH) λ_max_ (log *ε*) 250.0 (4.25); ^1^H NMR (600 MHz, CDCl_3_) data, see Table 5, ^13^C NMR (150 MHz, CDCl_3_) data, see Table 4; HRESIMS *m/z* 565.31287 [M + Na]^+^ (calcd for C_32_H_46_O_7_Na, 565.31412).

Ganoaustralenone L (**13**): yellow oil. [*α*]${}_{D}^{24}$ +44.67 (*c* 0.05, MeOH); UV (MeOH) λ_max_ (log *ε*) 250.0 (3.91); ^1^H NMR (500 MHz, CDCl_3_) data, see Table 5, ^13^C NMR (125 MHz, CDCl_3_) data, see Table 4; HRESIMS *m/z* 563.29688 [M + Na]^+^ (calcd for C_32_H_44_O_7_Na, 563.29792).

### 2.5. Synthesis of the Phenylglycine Methyl Ester (PGME) Derivatives

To a solution of **1** (2.0 mg, 3.9 μmol) in DMF (0.5 mL) on ice add PyBOP (2.5 mg, 4.8 μmol), HBTU (1.9 mg, 5.0 μmol), *N*-methylmorpholine 100 μL, and (*S*)-PGME (1.0 mg, 4.9 μmol). The reaction mixture was stirred at room temperature for 3 h. The reaction was stopped by adding 1 mL of EtOAc and then washed with H_2_O. The EtOAc layer was concentrated under reduced pressure to obtain a pale yellow oil sample, which was purified by HPLC to furnish (*S*)-PGME amide product **1a**. Similarly, (*R*)-PGME amide product **1b** was prepared from **1** (2.0 mg) and (*R*)-PGME (1.0 mg). NMR assignments of the protons for **1a** and **1b** were achieved by analysis of the ^1^H-^1^H COSY spectra.

Similarly, **2a** was prepared from **2** (0.5 mg) and (*S*)-PGME, **2b** was prepared from **2** (0.5 mg) and (*R*)-PGME. **3a** was prepared from **3** (1 mg) and (*S*)-PGME, **3b** was prepared from **3** (1 mg) and (*R*)-PGME. NMR assignments of the protons for **2a** and **2b** were achieved by analysis of the ^1^H-^1^H COSY spectra.

To a solution of **6** (2.5 mg) in THF (1 mL) was added 1 mL of LiOH (1 mol/L). The reaction mixture was stirred at room temperature overnight. The reaction was stopped by concentrating under reduced pressure to obtain a pale yellow oil sample, which was purified by HPLC to obtain **6H** (0.4 mg). Then **6Ha** was prepared from **6H** (0.2 mg) with (*S*)-PGME, **6Hb** was prepared from **6H** (0.2 mg) with (*R*)-PGME. NMR assignments of the protons for **6Ha** and **6Hb** were achieved by analysis of the ^1^H-^1^H COSY spectra.

**1a**: ^1^H NMR (600 MHz, CDCl_3_) 2.824 (1H, m, H-1a), 1.814 (1H, m, H-1b), 1.887 (1H, m H-2a), 2273 (1H, m, H-2b), 2.393 (1H, m, H-5), 2.247 (1H, m, H-6a), 2.530 (1H, m, H-6b), 2.480 (1H, m, H-12a), 2.770 (1H, m, H-12b), 2.500 (1H, m, H-15a), 2.923 (1H, m, H-15b), 2.608 (1H, m, H-16a), 1.737 (1H, m H-16b), 2.962 (1H, m, H-17), 0.667 (3H, s, H-18), 1.266 (3H, s, H-19), 2.141 (3H, s, H-21), 6.321 (1H, s, H-22), 4.123 (1H, dd, *J* = 6.0, 3.3 Hz, H-24), 2.924 (1H, m, H-25), 1.377 (3H, d, *J* = 7.2 Hz, H-27), 1.133 (3H, s, H-28), 1.108 (3H, s, H-29), 1.291 (1H, s, H-30), 6.892 (1H, d, *J* = 7.0 Hz, NH), 5.423 (1H, d, *J* = 7.0 Hz, H-2′ of PGME), 7.343–7.307 (5H, overlapped, phenyl group), 3.709 (3H, s, OCH_3_). HRESIMS *m/z* 682.33380 [M + Na]^+^ (calcd for C_39_H_49_O_8_NNa, 682.33559).

**1b**: ^1^H NMR (600 MHz, CDCl_3_) 2.818 (1H, m, H-1a), 1.855 (1H, m, H-1b), 1.910 (1H, m H-2a), 2265 (1H, m, H-2b), 2.388 (1H, m, H-5), 2.267 (1H, m, H-6a), 2.522 (1H, m, H-6b), 2.500 (1H, m, H-12a), 2.743 (1H, d, *J* = 16.6 Hz, H-12b), 2.515 (1H, m, H-15a), 2.927 (1H, m, H-15b), 2.613 (1H, m, H-16a), 1.744 (1H, m H-16b), 2.972 (1H, m, H-17), 0.536 (3H, s, H-18), 1.266 (3H, s, H-19), 2.098 (3H, s, H-21), 6.308 (1H, s, H-22), 4.132 (1H, dd, *J* = 6.0, 3.3 Hz, H-24), 2.927 (1H, m, H-25), 1.387 (3H, d, *J* = 7.2 Hz, H-27), 1.133 (3H, s, H-28), 1.110 (3H, s, H-29), 1.290 (1H, s, H-30), 7.061 (1H, d, *J* = 6.7 Hz, NH), 5.428 (1H, d, *J* = 6.7 Hz, H-2′ of PGME), 7.352–7.309 (5H, overlapped, phenyl group), 3.699 (3H, s, OCH_3_). HRESIMS *m/z* 682.33392 [M + Na]^+^ (calcd for C_39_H_49_O_8_NNa, 682.33559).

**2a**: ^1^H NMR (600 MHz, CDCl_3_) 2.782 (1H, m, H-1a), 1.838 (1H, m, H-1b), 2.324 (1H, m H-2a), 2720 (1H, m, H-2b), 2.297 (1H, d, *J* = 13.5, H-5), 4.408 (1H, dd, *J* = 13.5, 3.1 Hz, H-6), 2.505 (1H, d, *J* = 17.0 Hz, H-12a), 2.784 (1H, m, H-12b), 1.708 (1H, m, H-15a), 2.249 (1H, m, H-15b), 1.906 (1H, m, H-16a), 2.835 (1H, m H-16b), 1.932 (1H, m, H-17), 0.658 (3H, s, H-18), 1.219 (3H, s, H-19), 2.131 (3H, s, H-21), 6.328 (1H, s, H-22), 4.115 (1H, dd, *J* = 6.0, 3.4 Hz, H-24), 2.932 (1H, dd, *J* = 7.0, 3.4 Hz, H-25), 1.378 (3H, d, *J* = 7.0 Hz, H-27), 1.343 (3H, s, H-28), 1.438 (3H, s, H-29), 1.362 (1H, s, H-30), 6.875 (1H, d, *J* = 7.0 Hz, NH), 5.417 (1H, d, *J* = 6.8 Hz, H-2′ of PGME), 7.352–7.301 (5H, overlapped, phenyl group), 3.704 (3H, s, OCH_3_). HRESIMS *m/z* 698.33087 [M + Na]^+^ (calcd for C_39_H_49_O_9_NNa, 698.33050).

**2b**: ^1^H NMR (600 MHz, CDCl_3_) 2.873 (1H, m, H-1a), 1.838 (1H, m, H-1b), 2.328 (1H, m H-2a), 2722 (1H, m, H-2b), 2.293 (1H, d, *J* = 13.5, H-5), 4.400 (1H, dd, *J* = 13.5, 3.1 Hz, H-6), 2.399 (1H, d, *J* = 17.0 Hz, H-12a), 2.754 (1H, m, H-12b), 1.700 (1H, m, H-15a), 2.243 (1H, m, H-15b), 1.906 (1H, m, H-16a), 2.839 (1H, m H-16b), 1.958 (1H, m, H-17), 0.519 (3H, s, H-18), 1.221 (3H, s, H-19), 2.093 (3H, s, H-21), 6.315 (1H, s, H-22), 4.128 (1H, dd, *J* = 5.0, 3.5 Hz, H-24), 2.935 (1H, dd, *J* = 7.3, 3.5 Hz, H-25), 1.386 (3H, d, *J* = 7.2 Hz, H-27), 1.343 (3H, s, H-28), 1.440 (3H, s, H-29), 1.348 (1H, s, H-30), 7.047 (1H, d, *J* = 6.6 Hz, NH), 5.419 (1H, d, *J* = 6.6 Hz, H-2′ of PGME), 7.347–7.277 (5H, overlapped, phenyl group), 3.696 (3H, s, OCH_3_). HRESIMS *m/z* 698.32990 [M + Na]^+^ (calcd for C_39_H_49_O_9_NNa, 698.33050).

**3a**: ^1^H NMR (600 MHz, CDCl_3_) 2.976 (1H, m, H-1a), 1.709 (1H, m, H-1b), 2.474 (1H, m H-2a), 2595 (1H, m, H-2b), 2.084 (1H, overlapped H-5), 2.902 (1H, overlapped, H-6a), 1.687 (1H, overlapped, H-6b), 4.459 (1H, overlapped, H-7), 2.749 (1H, d, *J* = 17.3 Hz, H-12a), 2.375 (1H, d, *J* = 17.3 Hz, H-12b), 1.701 (1H, m, H-15a), 2.439 (1H, m, H-15b), 1.978 (1H, m, H-16a), 2.026 (1H, m H-16b), 2.981 (1H, m, H-17), 0.642 (3H, s, H-18), 1.022 (3H, s, H-19), 2.137 (3H, s, H-21), 6.343 (1H, s, H-22), 4.137 (1H, dd, *J* = 6.0, 3.4 Hz, H-24), 2.939 (1H, dd, *J* = 7.3, 3.4 Hz, H-25), 1.385 (3H, d, *J* = 7.3 Hz, H-27), 1.147 (3H, s, H-28), 1.073 (3H, s, H-29), 1.344 (1H, s, H-30), 6.903 (1H, d, *J* = 6.9 Hz, NH), 5.426 (1H, d, *J* = 6.9 Hz, H-2′ of PGME), 7.343–7.306 (5H, overlapped, phenyl group), 3.703 (3H, s, OCH_3_). HRESIMS *m/z* 684.35034 [M + Na]^+^ (calcd for C_39_H_51_O_8_NNa, 684.35124).

**3b**: ^1^H NMR (600 MHz, CDCl_3_) 2.976 (1H, m, H-1a), 1.692 (1H, m, H-1b), 2.609 (1H, m H-2a), 2.458 (1H, m, H-2b), 2.073 (1H, overlapped H-5), 1.691 (1H, overlapped, H-6a), 1.638 (1H, overlapped, H-6b), 4.438 (1H, t, *J* = 4.2 Hz, H-7), 2.711 (1H, d, *J* = 16.9 Hz, H-12a), 2.692 (1H, d, *J* = 16.9 Hz, H-12b), 1.809 (1H, m, H-15a), 2.021 (1H, m, H-15b), 1.977 (1H, m, H-16a), 1.911 (1H, m H-16b), 2.904 (1H, m, H-17), 0.463 (3H, s, H-18), 1.025 (3H, s, H-19), 2.096 (3H, s, H-21), 6.316 (1H, s, H-22), 4.136 (1H, dd, *J* = 4.5, 3.7 Hz, H-24), 2.948 (1H, dd, *J* = 7.3, 3.7 Hz, H-25), 1.402 (3H, d, *J* = 7.3 Hz, H-27), 1.145 (3H, s, H-28), 1.076 (3H, s, H-29), 1.326 (1H, s, H-30), 7.116 (1H, d, *J* = 6.9 Hz, NH), 5.420 (1H, d, *J* = 6.9 Hz, H-2′ of PGME), 7.333–7.265 (5H, overlapped, phenyl group), 3.697 (3H, s, OCH_3_). HRESIMS *m/z* 684.35022 [M + Na]^+^ (calcd for C_39_H_51_O_8_NNa, 684.35124).

**6Ha**: ^1^H NMR (600 MHz, CDCl_3_) 2.260 (1H, m, H-1a), 2.015 (1H, m, H-1b), 2.953 (1H, m H-2a), 2.347 (1H, m, H-2b), 1.861 (1H, overlapped H-5), 2.300 (1H, overlapped, H-6a), 2.295 (1H, overlapped, H-6b), 6.510 (1H, overlapped, H-7), 5.666 (1H, overlapped, H-11), 4.585 (1H, m, H-15), 2.500 (1H, m, H-16a), 1.812 (1H, m H-16b), 3.308 (1H, m, H-17), 0.907 (3H, s, H-18), 1.095 (3H, s, H-19), 2.273 (3H, s, H-21), 6.365 (1H, s, H-22), 2.540 (1H, m H-24a), 2.540 (1H, m H-24b), 2.937 (1H, m, H-25), 1.139 (3H, d, *J* = 6.8 Hz, H-27), 1.122 (3H, s, H-28), 1.161 (3H, s, H-29), 1.298 (1H, s, H-30), 6.914 (1H, d, *J* = 7.0 Hz, NH), 5.487 (1H, d, *J* = 7.0 Hz, H-2′ of PGME), 7.355–7.338 (5H, overlapped, phenyl group), 3.686 (3H, s, OCH_3_). HRESIMS *m/z* 666.33997 [M + Na]^+^ (calcd for C_39_H_49_O_7_NNa, 666.34067).

**6Hb**: ^1^H NMR (600 MHz, CDCl_3_) 2.267 (1H, m, H-1a), 1.790 (1H, m, H-1b), 2.790 (1H, m H-2a), 2.875 (1H, m, H-2b), 1.875 (1H, overlapped H-5), 2.286 (1H, overlapped, H-6a), 2.301 (1H, overlapped, H-6b), 6.494 (1H, overlapped, H-7), 5.662 (1H, s, H-11), 4.553 (1H, m, H-15), 2.453 (1H, m, H-16a), 1.477 (1H, m H-16b), 3.267 (1H, m, H-17), 1.295 (3H, s, H-18), 1.080 (3H, s, H-19), 2.197 (3H, s, H-21), 6.289 (1H, s, H-22), 2.510 (1H, m H-24a), 2.510 (1H, m H-24b), 2.914 (1H, m, H-25), 1.200 (3H, d, *J* = 7.2 Hz, H-27), 1.120 (3H, s, H-28), 1.160 (3H, s, H-29), 0.821 (1H, s, H-30), 6.803 (1H, d, *J* = 7.2 Hz, NH), 5.516 (1H, d, *J* = 7.2 Hz, H-2′ of PGME), 7.360–7.293 (5H, overlapped, phenyl group), 3.722 (3H, s, OCH_3_). HRESIMS *m/z* 666.33978 [M + Na]^+^ (calcd for C_39_H_49_O_7_NNa, 666.34067).

### 2.6. Biological Activity Assays

Biological activity assays, including the cytotoxicity against five human cancer cell lines [24], α-glucosidase inhibition [25], protein tyrosine phosphatase 1 β (PTP1B) [26], dipeptidyl peptidase 4 (DDP4) [27], and angiotensin converting enzyme 2 (ACE2) [28], were screened according to the protocols in the Appendix A.

## 3. Results and Discussion

Compound **1** (Figure 1), obtained as a yellow oil, gave an [M + Na]^+^ ion peak at *m/z* 535.26685 in the HRESIMS (calcd for C_30_H_40_O_7_Na, 535.26717). The ^1^H NMR spectroscopic data (Table 1) displayed six methyl singlets at *δ*_H_ 1.28 (Me-19), *δ*_H_ 0.75 (Me-18), *δ*_H_ 2.16 (Me-21), *δ*_H_ 1.13 (Me-28), *δ*_H_ 1.10 (Me-29), and *δ*_H_ 1.35 (Me-30), one methyl doublet at *δ*_H_ 1.18 (d, *J* = 7.3 Hz, Me-27), an olefinic proton at *δ*_H_ 6.51 (s, H-22), and an oxygenated methine proton at *δ*_H_ 4.21 (d, *J* = 4.9 Hz, H-24). The ^13^C NMR and DEPT spectroscopic data (Table 2) of **1** showed 30 carbon resonances which were ascribed to seven methyl carbons at *δ*_C_ 19.1 (C-18), 18.4 (C-19), 22.2 (C-21), 13.8 (C-27), 27.7 (C-28), 20.7 (C-29), and 26.4 (C-30), six methylenes at *δ*_C_ 36.1 (C-1), 35.0 (C-2), 37.7 (C-6), 50.9 (C-12), 33.5 (C-15), and 24.2 (C-16), four methines at *δ*_C_ 50.8 (C-5), 54.6 (C-17), 88.0 (C-24), and 44.6 (C-25), two pairs of olefinic carbons at *δ*_C_ 151.3 (C-8), 152.4 (C-9), 161.3 (C-20), and 122.0 (C-22), four *sp^3^*-quaternary carbons at *δ*_C_ 47.8 (C-4), 40.1 (C-10), 49.7 (C-13, C-14), and four carbonyls at *δ*_C_ 218.6 (C-3), 202.7 (C-7), 202.8 (C-11), 202.0 (C-23), and 176.9 (C-27). The chemical shifts of 1D NMR of **1** indicated that it was a lanostane triterpenoid similar to resinacein N, except for the substitutions at C-3, C-7, and C-15 [29]. In the HMBC spectrum of **1**, the correlations from Me-29 to the carbonyl C-3, from Me-30 to the methylene carbon C-15, and from H-5 (*δ*_H_ 2.38) and H-6 (*δ*_H_ 2.36, 2.68) to the carbonyl C-7, along with the ^1^H-^1^H COSY correlation of H-15 (*δ*_H_ 2.24, 1.87)/H-16 (*δ*_H_ 2.04, 1.91) (Figure 2), suggested that C-3 and C-7 were ketone carbons and C-15 was a methylene instead of being a hydroxylated methine in resinacein N. Therefore, the planar structure of **1** was elucidated as shown in Figure 1.

The key ROESY correlations between H-22 (*δ*_H_ 6.51) and H-16a/b (*δ*_H_ 2.04, 1.91) allowed the assignment of the *E* configuration of the C-20–C-22 double bond (Figure 3). The absolute configuration of the chiral center C-25 was determined by the PGME method (Figure 4). The (*R*)- and (*S*)-PGME amide derivatives were chemically synthesized, and the Δ*δ*_H_ (*δ*_S_ − *δ*_R_) values indicated that C-25 was the *S* configuration. The attempt to assign the absolute configuration of C-24 by Mosher’s method failed, probably due to the bulky groups around the hydroxy group. Therefore, the configuration of C-24 remained unassigned. Compound **1** was elucidated as [20(22)*E*,24*R*,25*R*]-24-hydroxy-3,7,11,23-tetraoxolanosta-8,20-dien-26-oic acid, and was given the trivial name ganoaustralenone A.

Compound **2**, obtained as a white powder, displayed an [M + Na]^+^ peak at *m/z* 551.26422 in the HRESIMS (calcd for C_30_H_40_O_8_Na, 551.26209). The 1D NMR data of **2** (Table 1 and Table 2) showed a resemblance to those of compound **1**, implying the analogous structures of the two compounds. Analysis of the 1D NMR data suggested that the only difference between **1** and **2** was C-6. The HMBC correlation from H-5 to a hydroxymethine at *δ*_C_ 72.2 (C-6), as well as the ^1^H-^1^H COSY correlation of H-5 and the proton at *δ*_H_ 4.44 (H-6) (Figure 2), revealed that the C-6 in **2** attached to a hydroxy group compared to that of **1**. These assignments are consistent with the HRESIMS result. The absolute configuration of C-25 was determined by the PGME method, as in the case of compound **1** (Figure 4). Therefore, compound **2** was determined as shown in Figure 1, and trivially named ganoaustralenone B.

Compound **3**, obtained as a yellow oil, displayed an [M + Na]^+^ ion peak at *m/z* 537.28180 in the HRESIMS analysis (calcd for C_30_H_42_O_7_Na, 537.28227). The ^1^H NMR and ^13^C NMR data of **3** (Table 1 and Table 2) highly resemble those of **1**, except for the chemical shift of C-7. The key ^1^H-^1^H COSY correlations H-5 (*δ*_H_ 2.11)/H-6 (*δ*_H_ 1.70)/H-7 (*δ*_H_ 4.47), as well as the HMBC correlation from H-7 (*δ*_H_ 4.47) and C-8 (*δ*_C_ 160.3) (Figure 2), implied the presence of a hydroxyl group at C-7. The key ROESY correlations of H-7/H-15β/H_3_-18 indicated the α orientation of 7-OH (Figure 3). The absolute configurations of C-25 were determined by the PGME method, as in the case of compound **1** (Figure 4). Therefore, compound **3** was determined as shown in Figure 1, and identified as ganoaustralenone C.

The yellowish oil compounds **4** and **5** gave the sodium adduct ion peaks of *m/z* 565.27728 and *m/z* 549.28210 in the HRESIMS analysis, corresponding to the molecular formulas of C_31_H_42_O_8_ and C_31_H_42_O_7_ (calcd for C_31_H_42_O_8_Na 565.27774; C_31_H_42_O_7_Na, 549.28282), respectively. The 1D NMR spectra of the two compounds (Table 1 and Table 2) showed characteristic signals of triterpene, indicating the same skeletons of **1**–**5**. Analysis of the 1D NMR spectra of **4** and **5** suggested that the two compounds were highly similar to those of **1** and **2**, respectively. The differences between these two pairs of compounds (**1** vs. **4**, **2** vs. **5**) were the status of C-26 carboxylic group. The correlations from the methoxy singlets to the carbonyl group (C-26) in the HMBC spectra of **4** and **5** (Figure 2) indicated that C-26 of **4** and **5** have been methyl esterified instead of being free carboxylic groups in **1** and **2**. Therefore, compounds **4** and **5** were elucidated as the C-26 methyl ester derivatives of **1** and **2**, respectively. However, these changes hampered the absolute configuration determination of C-25 of **4** and **5** by the PGME method. The relative configurations of C-24 and C-25 were assigned as *R** and *S**, respectively, by analysis of the Newman projections of C-24–C-25 and the coupling constants of H-24 (4.0 Hz, and 3.1 Hz). Therefore, compounds **4** and **5** were named ganoaustralenones D and E, respectively.

Compound **6** was determined to be methyl gibbosate O by comparison with the NMR spectroscopic data (Appendix A) [30,31]. However, the chemical shifts of C-13 and C-14 of methyl gibbosate O have been erroneously assigned previously [30]. The key HMBC correlation of H-11 (*δ*_H_ 5.66, s) to an *sp^3^*-quaternary carbon at *δ*_C_ 58.0, together with the HMBC correlation from H-7 (*δ*_H_ 6.50, m) to an *sp^3^*-quaternary carbon at *δ*_C_ 52.5 enabled the correct assignment of the chemical shifts of C-13 (58.0 ppm) and C-14 (52.5 ppm). Moreover, the absolute configuration of C-25 of gibbosic acid O was assigned as *S* without any evidence [31], while for methyl gibbosate O, the C-25 configuration was assigned to be same with gibbosic acid O only by comparison with the chemical shifts [30]. However, C-25 is far away from any other chiral centers in the structure, so the chemical shift deviation is inadequate to discriminate the *S* and *R* configuration of C-25. Therefore, more solid evidence should be presented to corroborate the real configuration of C-25. In order to determine the absolute configuration of the chiral center C-25, compound **6** was firstly hydrolyzed by LiOH to obtain the previously reported compound gibbosic acid O (**6H**). Then, the (*R*)- and (*S*)-PGME amide derivatives of **6H** were chemically synthesized (Figure 1), and the Δ*δ*_H_ (*δ*_S_ − *δ*_R_) values indicated that C-25 was the *S* configuration (Figure 4). Therefore, the absolute configuration of compound **6** has been fully assigned.

The HRESIMS analysis of **7**, a yellow oily compound, gave a sodium adduct ion peak at *m/z* 547.30286, corresponding to the molecular formula of C_32_H_44_O_6_ (calcd for C_32_H_44_O_6_Na, 547.30356) with 11 double bond equivalences. Comparing the 1D NMR data of **7** (Table 2 and Table 3) with those of **6** suggested that **7** differed from **6** with the presence of an oxygenated methylene and a triplet methyl group with the absence of the methoxy group. These signals were assigned to be ethyl ester moiety of the C-26 carbonyl group instead of the methyl ester moiety in **6**. The ^1^H-^1^H COSY correlation of OCH_2_CH_3_ (*δ*_H_ 4.13)/OCH_2_CH_3_ (*δ*_H_ 1.25), and the HMBC correlation from OCH_2_CH_3_ (*δ*_H_ 4.13) to C-27 (*δ*_C_ 176.3) (Figure 2), confirmed the above assignments. Notably, 15-OH was assigned to be *β* orientation by the key ROESY correlation of H-15 (*δ*_H_ 4.31)/Me-30 (*δ*_H_ 1.00) (Figure 3). Therefore, compound **7** was named ganoaustralenone F.

Compound **8** had an [M + Na]^+^ ion peak at *m/z* 561.28223 (C_32_H_42_O_7_Na) in the HRESIMS analysis (calcd for C_32_H_42_O_7_Na, 561.28282). The molecular formula of **8** is two oxygen atoms more than that of **7**, indicating the existence of more oxygenated carbons in **8** than those of **7**. The ^1^H NMR spectra of **8** (Table 3) displayed six methyl singlets (*δ*_H_ 1.12, 1.08, 2.26, 1.12, 1.09, and 1.29). The ^13^C NMR (Table 4) and DEPT spectra of **8** exhibited signals for eight methyls, six methylenes (one was oxygenated at *δ*_C_ 60.7), six methines including two *sp*^2^-ones and four *sp^3^*-ones (one was oxygenated, *δ*_C_ 58.9), and eleven quaternary carbons (four carbonyls, five *sp^3^*-ones, and two *sp^2^*-ones). Further analysis of the 1D NMR data (Table 3 and Table 4) allowed the assignment of **8** to be an analog of **7**, except for the positions of C-7, C-8, and C-15. The ^13^C NMR chemical shifts of these three positions (*δ*_C_ 58.9, C-7; *δ*_C_ 62.7, C-8; *δ*_C_ 209.9, C-15) of **8** implied that an epoxy ring was located at C-7 and C-8, while C-15 was a ketone compared to that of **7**. These assignments were corroborated by the ^1^H-^1^H COSY correlations of H-5/H-6/H-7 and the HMBC correlations from H-7 to C-8, and from Me-30 to C-15 (Figure 2). Thus, compound **8** was trivially named ganoaustralenone G.

Compound **9** showed an [M + H]^+^ peak at *m/z* 513.32135 in the HRESIMS, indicating the molecular formula C_31_H_44_O_6_ (calcd for C_31_H_45_O_6_, 513.32161). The 1D NMR data of **9** (Table 3 and Table 4) displayed thirty-one carbon resonances, which were categorized into seven methyl carbons at *δ*_C_ 17.7 (C-19), 18.6 (C-18), 21.4 (C-21), 17.3 (C-27), 27.8 (C-28), 20.6 (C-29), and 27.8 (C-30), one methoxy carbon at *δ*_C_ 52.0, seven methylenes at *δ*_C_ 34.2 (C-1), 34.9 (C-2), 29.4 (C-6), 50.3 (C-12), 30.4 (C-15), 23.2 (C-16), and 47.9 (C-24), five methines at *δ*_C_ 45.2 (C-5), 67.3 (C-7), 54.1 (C-17), 123.8 (C-22), and 35.0 (C-25), and eleven proton-free carbons at *δ*_C_ 218.0 (C-3), 46.5 (C-4), 159.5 (C-8), 140.2 (C-9), 38.0 (C-10), 195.5 (C-11), 48.6 (C-13), 50.6 (C-14), 158.0 (C-20), 198.4 (C-23), and 176.7 (C-26). The above data suggested that **9** was a similar structure to 7β-hydroxy-3,11,15,23-tetraoxolanosta-8,20*E*(22)-diene-26-oic acid methyl ester, except for the position at C-15 and the configuration of 7-OH [32]. The ^1^H-^1^H COSY correlations of H-15/H-16/H-17, as well as the HMBC correlation from Me-30 (*δ*_H_ 1.35, s) to the methylene carbon at *δ*_C_ 30.4 (C-15) (Figure 2), suggested that C-15 in **9** was a methylene instead of being a ketone carbon in 7β-hydroxy-3,11,15,23-tetraoxolanosta-8,20*E*(22)-diene-26-oic acid methyl ester. In addition, the key ROESY correlations of H-7 (*δ*_H_ 4.47)/H-15β (*δ*_H_ 2.04)/Me-18 (*δ*_H_ 0.67) (Figure 3) suggested that 7-OH was an *α* orientation. Therefore, compound **9** was named ganoaustralenone H.

Compound **10**, a pale yellow oil, gave an [M + Na]^+^ ion peak at *m/z* 563.29749 (C_32_H_44_O_7_Na) in the HRESIMS (calcd for C_32_H_44_O_7_Na, 563.29847). The ^1^H and ^13^C NMR spectroscopic data of **10** (Table 3 and Table 4) showed high similarity to those of the structure 15α-hydroxy-3,11,23-trioxolanosta-8,20*E*(22)-dien-26-oic acid methyl ester, a lanostane triterpenoid isolated from the *G. lucidum* [33]. Further analysis of the 2D NMR spectra revealed that the only difference between these two structures was C-7. The diagnostic HMBC correlations from the protons at *δ*_H_ 2.26 (H-5), 2.49 (H-6a), 2.62 (H-6b) to a carbonyl group at *δ*_C_ 204.4 (Figure 2) suggested that C-7 was a carbonyl group in **10** instead of being a methylene group in 15α-hydroxy-3,11,23-trioxolanosta-8,20*E*(22)-dien-26-oic acid methyl ester. In addition, the alcohol for forming the C-26 ester group was ethanol in **10** instead of methanol in the reported structure, as supported by the two chemical shifts at *δ*_C_ 60.6 (-OCH_2_CH_3_) and 14.2 (-OCH_2_CH_3_). Therefore, compound **10** was identified as ethyl 20(22)*E*-15α-hydroxy-3,7,11,23-tetraoxolanosta-8,20(22)-dien-26-oate, and was trivially named ganoaustralenone I.

Compound **11**, obtained as a yellow oil, displayed an [M + Na]^+^ ion peak at *m/z* 549.28210 in the HRESIMS analysis (calcd for C_31_H_42_O_7_Na, 549.28282), revealing the molecular formula of C_31_H_42_O_7_. The 1D NMR data of **11** (Table 4 and Table 5) showed 30 carbon resonances with high resemblance to those of compound **10**. Further analysis of the 2D NMR data (Figure 2 and Figure 3) suggested that **11** differed from **10** by the presence of the methyl ester group. The significant HMBC correlation from the methoxy group (*δ*_H_ 3.68) to the carbonyl group C-26 (*δ*_C_ 176.6) (Figure 2) verified the terminal carboxylic group in **11** has been methyl esterified instead of being ethyl esterified in **10**. Therefore, compound **11** was identified as ganoaustralenone J.

The pale yellow oil compound **12** exhibited an [M + Na]^+^ ion peak at *m/z* 565.31287 in the HRESIMS analysis (calcd for C_32_H_46_O_7_Na, 565.31412). The NMR spectroscopic data of **12** (Table 4 and Table 5) highly resemble those of **9**, except for the chemical shifts of C-24 and the alcoholic part of the C-26 ester. The important HMBC correlations from Me-27 (*δ*_H_ 1.30) to a hydroxymethine at *δ*_C_ 78.6 (C-24) (Figure 2), together with the chemical shifts of the alcoholic part [*δ*_C_ 61.0 (-OCH_2_CH_3_) and 14.3 (-OCH_2_CH_3_)], indicated that a hydroxy group situated at C-24 and the presence of ethyl ester of C-26 in **12** compared to those of **9**. Therefore, compound **12** was identified as ganoaustralenone K.

Compound **13**, obtained as a yellow oil, displayed an [M + Na]^+^ ion peak at *m/z* 563.29688 in the HRESIMS analysis (calcd for C_32_H_44_O_7_Na, 563.29792). Analysis of the ^1^H and ^13^C NMR data (Table 4 and Table 5) revealed that this compound was a structural analog to **12**. The main difference between the NMR data of the two analogs was the position C-7 (*δ*_C_ 201.2), which indicated that C-7 was a carbonyl carbon. In the HMBC spectrum of compound **13**, significant correlations from H_2_-6 (*δ*_H_ 2.48, 2.34) to C-7 (*δ*_C_ 201.2) (Figure 2) indicated that C-7 was a carbonyl carbon. Therefore compound **13** was identified as ganoaustralenone L.

The identification of a series of 20(22)*E*-lanostanes from this species of *Ganoderma* inspired a proposal of the possible biosynthetic pathways. Take compound **1** as an example, as shown in Figure 2, the common precursor squalene, which was derived from two molecules of farnesyl pyrophosphate, which was oxygenated and followed by function migration to give the lanostane scaffold. The lanosterol was oxygenated at the positions of C-3, C-7, C-11, C-20, C-23, and C-26 to give the key intermediate **A**, which underwent an elimination reaction by the E1cb mechanism to yield **B**. Finally, a nucleophilic attack at C-24 by a hydroxy group produced compound **1**.

All the isolates were subjected to evaluate their cytotoxicity against the five human cancer cell lines (the HL-60 (ATCC CCL-240), the human myeloid leukemia cell line, the SMMC-7721 human hepatocellular carcinoma cell line, the A549 (ATCC CCL-185) lung cancer cell line, the MCF-7 (ATCC HTB-22) breast cancer cell line, and the SW-480 (ATCC CCL-228) human colon cancer cell line, as well as the inhibitory activity against α-glucosidase, protein tyrosine phosphate 1 β (PTP1B), dipeptidyl peptidase 4 (DDP4), and angiotensin-converting enzyme 2 (ACE2). However, no significant bioactivity was observed.

## 4. Conclusions

In conclusion, twelve previously undescribed lanostane-type triterpenes were obtained from the medicinal mushroom *Ganoderma australe*. By using the NMR and HRESIMS techniques for structural elucidation, the structures of twelve triterpenes were determined, and the absolute configurations of **1**, **2,** and **6** were assigned by the phenylglycine methyl ester (PGME) method. Ganoderma triterpenes have been reported to have more than 400 chemical entities to date [15]. Most of them were oxygenated at the positions of C-3, C-7, C-11, C-15, and C-26. Interestingly, more and more studies have revealed that there was an oxygenated position bias that differed from species to species. The triterpenes described here are featured by an unusual 20(22)-*trans* double bond, which has rarely been found in the *Ganoderma* lanostanoid family. Although no significant biological activities were found in this study, the results have also initiated the understanding of the structural diversity of *Ganoderma*-derived triterpenoids.

## Data Availability

The data presented in this study are available in this manuscript and can be requested from the corresponding author.

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
