# Peer review of "[20(22)E]-Lanostane Triterpenes from the Fungus Ganoderma australe"

_jof, 2022, doi:10.3390/jof8050503_

Round 1
Reviewer 1 Report
This manuscript of Lin Zhou et al. devoted to isolation and structure elucidation of Ganoderma australe metabolites, basidiomycete fungi used in Chinees traditional medicine. Described study is good and interesting, but there are some questions for the authors.
First of all, because you often use methanol in your work, could do you explain, whether compounds 3-13 can be natural metabolites, and not extraction artifacts?
Next, why you had not established absolute configurations of C-24 and C-25 in compound 3? It's obvious, that the PGME method applicable in this case. And why you had not established absolute configurations of C-24 and C-25 in 4-13? Demethylation is a very simple reaction (and safe for other moieties in molecule). About absolute configurations of C-25 in 6 you have wrote, that "The absolute configuration of C-25 remained unassigned due to the lack of enough samples for the preparation of PGME derivatives". But compound 6 was isolated in amount of 5.1 mg, however in case of compounds 4 and 5 you have wrote, that "compounds 4 and 5 were elucidated as the C-26 methyl ester derivatives of 1 and 2, respectively. However, these changes hampered the absolute configuration determination of C-25 of 4 and 5 by PGME method". I think, the amount of compounds 4-6 is enough.
Also, I woud like dear authors, have you thought about taking CD spectra for compounds 3-13? It in complex with quantum-mechanical calculations may help to determine absolute configurations of C-24 and C-25, I suppose. In addition, other configurations of asymmetric centers are already known, which makes the task easier.
And, have you thought about using the Mosher's method to establish absolute configuration of C-24 in compounds 3-5, 12 and 13?
And, finally, what about bioactivity of isolated compounds? In the introduction, you state that this fungi has healing properties, however, this is not clear from the your article.
Please, find other detailed corrections in attached file.

Reviewer 2 Report
The authors describe the isolation of 13 compounds that have minimal differences among themselves. Substances were isolated from the fungus Ganoderma australe. In general, the manuscript looks interesting, but there are a number of comments. The introduction is very short and there is no discussion at all. If the authors do not study the biological activity of isolated compounds (by the way, why?), then they need to shed more light on the issue of the origin of these metabolites and propose a biosynthesis scheme. In addition, it is necessary to consider in detail the structural similarity of the isolated metabolites with others isolated from closely related species. In the discussion, it is necessary to describe the diversity of lanostane triterpenoids and what is the uniqueness of those identified by the authors.
Reviewer 3 Report
The article [20(22)E]-Lanostane Triterpenes from the Fungus Ganoderma australe by authors Lin Zhou , Li-Li Guo , Masahiko Isaka , Zheng-Hui Li , He-Ping Chen presents the isolation of 13 new ganoderic acid derivatives of the Ganoderma australe fungus.
The structural discernment work is very exhaustive and detailed. The compounds show a high structural similarity, and the authors have worked efficiently in two-dimensional methods to achieve an thorough assignment of each signal. The same as in the separative techniques.
I consider that the work is novel and presents an adequate experimental and theoretical work, as a result of which the isolation of the presented compounds was achieved.
The minimum improvements that the work should have come from the large amount of information that it handles, which makes its organization difficult and that is translated into the manuscript. The tables must be reorganized before publication, mainly Table 4.
Round 2
Reviewer 1 Report
Dear authors!
First of all I wanna thank the authors for detailed answers and detailed explanations. Also I wanna note the authors added some explanations to the article that help to readers better understand this study. But there are some things that can be improved.
Please, pay attention to red marks in figure 4 (please, find attached file). This legends not corresponds to reality. Herein depicted the ΔσH between σS and σR of PGME derivatives of 1, 2, 3 and 6. But in your case this legend indicates to depicted the S-PGME and R-PGME derivatives for each ones.
About bioactivities. Why did you choose these particular tests for your compounds? Why don't you try testing these lanostane triterpenoids for anti-inflammatory activity? By the consumption of compounds. Do you have a possibility to collect new samples of Ganoderma australe? This would help you isolate the additional quantities of required compounds, since the isolation scheme is already known.
Also need to add your results of bioactivity investigations. Even if you did not get significant results, you still need to report it in the article.
About plausible scheme of biosynthesis. Need to add the structure of 1 in scheme. Also, please, add the groups of enzymes participating in this transformations.

Reviewer 2 Report
1. The authors significantly changed the manuscript, including revisiting the novelty of one compound that had already been isolated. But confusion remains in the text about the number of new compounds. In some places there are twelve new compounds, and in some places of the text there are still thirteen. This needs to be corrected. Also, the location of the structure of the known compound 6 among the new ones in Figures 1 and 2 and in the description is misleading that substance 6 is also new. It is necessary to correct this so as not to mislead the reviewer.
2. The isolated substances were examined for a variety of biological activities, but, surprisingly, no activity was found, despite the fact that the Ganoderma is a source of substances valuable in folk medicine. This seems contradictory to me. Lanostane triterpenoids are known biologically active compounds. There are a large number of publications on their activity, and the authors should be guided by these literature data when choosing bioassays in order to really get a result.
The resulting negative data should also be described so that other studies can build on them. Data on the cell lines used and tests with them, as well as experiments with enzymatic activity, should be given at least in an additional file.
So, I repeat that from the point of view of the medicinal value of the Ganoderma mushroom, experiments to study the biological activity of the isolated substances are inadequate. And in this sense, the value of the obtained information about the metabolites of the fungus is not high.
Round 3
Reviewer 2 Report
The authors have made possible changes, but nevertheless, I continue to believe that the study was poorly designed. Indeed, lanostane triterpenoids are a very large group of compounds and are not required to exhibit properties beneficial to humans. It is possible that these compounds have some kind of signaling effect for fungal and soil communities. Nevertheless, my experience in studying the biological activity of natural substances suggests that a careful preliminary analysis of the literature allows us to guess the result with a high degree of success. In addition, the use of cellular models rather than specific enzymes provides even better results. Thus, I continue to believe that the authors have done a good chemistry job, but it doesn't look good enough.
This manuscript is a resubmission of an earlier submission. The following is a list of the peer review reports and author responses from that submission.